# DIFFUSION-STEGO: TRAINING-FREE DIFFUSION GENERATIVE STEGANOGRAPHY VIA MESSAGE PROJECTION

## ABSTRACT

Generative steganography is the process of hiding secret messages in generated images instead of cover images. Existing studies on generative steganography use GAN or Flow models to obtain high hiding message capacity and anti-detection ability over cover images. However, they create relatively unrealistic stego images because of the inherent limitations of generative models. We propose Diffusion-Stego, a generative steganography approach based on diffusion models that outperform other generative models in image generation. Diffusion-Stego projects secret messages into the latent noise of diffusion models and generates stego images with an iterative denoising process. Since the naive hiding of secret messages into noise boosts visual degradation and decreases extracted message accuracy, we introduce message projection, which hides messages into noise space while addressing these issues. We suggest three options for message projection to adjust the trade-off between extracted message accuracy, anti-detection ability, and image quality. Diffusion-Stego is a training-free approach, so we can apply it to pre-trained diffusion models that generate high-quality images, or even large-scale text-to-image models, such as Stable diffusion. Diffusion-Stego achieved a high capacity of messages (3.0 bpp of binary messages with 98% accuracy, and 6.0 bpp with 90% accuracy) as well as high quality (with a FID score of 2.77 for 1.0 bpp on the FFHQ $64{\times}64$ dataset) that makes it challenging to distinguish from real images in the PNG format.

## 1 INTRODUCTION

Image steganography is the process that aims at hiding secret messages in images so that the secret messages are not detected or exposed by third-party players. Traditional image steganography methods (Morkel et al., 2005; Johnson & Jajodia, 1998) conceal secret messages within a natural cover image. The sender transmits the cover image containing the secret messages, termed a stego image, to the receiver, who extracts the hidden messages from the stego image. On the contrary, the third-party players attempt to discriminate the stego images by training steganalyzer models (Xu et al., 2016; Ye et al., 2017; Fridrich & Kodovsky, 2012), which classify over the cover images and stego images.

Generative steganography methods (Wu & Wang, 2014; Hu et al., 2018) have been proposed to deceive steganalyzer models. These approaches apply deep generative models that synthesize stego images from secret messages without using cover images. It makes them less vulnerable to steganalyzer models, as there are no cover images for the steganalyzer to train on. Recent generative steganography studies (Wei et al., 2022a; Zhou et al., 2022; Wei et al., 2022b) using Generative Adversarial Networks (GAN) (Goodfellow et al., 2020) or Flow (Kingma & Dhariwal, 2018) models as a generator have been proposed. In contrast to their anti-detection ability and high hiding capacity, they relatively lack image fidelity due to the limitations of the generative models they use.

Therefore, we explore utilizing diffusion-based generative models for steganography. Diffusion models (Sohl-Dickstein et al., 2015; Ho et al., 2020) are recently popular generative models, which generate high-quality images (Ramesh et al., 2022; Saharia et al., 2022; Rombach et al., 2022) with an iterative sampling process. Recent studies (Karras et al., 2022; Xu et al., 2023) have utilized diffu-

sion models and deterministic samplers (Song et al., 2021; 2020; Karras et al., 2022) for generating high quality images.

In this paper, we propose Diffusion-Stego, a powerful generative steganography approach utilizing diffusion models and deterministic sampler. As illustrated in Figure 1(a), Diffusion-Stego hides secret messages in noise and generates stego images using the deterministic sampler without re-training the diffusion models. Due to the invertible property of the deterministic sampler, Diffusion-Stego can generate and extract messages using a single diffusion model. This allows the sender and the receiver to share only a diffusion model and a hiding method.

However, we have identified that there are two challenges to utilizing a diffusion model in steganography. First, diffusion models cannot generate images from noise replaced with binary messages, which is the naive approach of hiding messages in the noise. Second, the accumulation of slight errors during the reverse process of the deterministic sampler leads to a drop in the extracted message accuracy. To address these challenges, we propose a novel technique called *message projection*. Message projection modifies noise to the extent that it does not deviate from the distribution of random noise while preserving the quality of stego images and ensuring high extracted message accuracy. We offer three types of message projection, which can be adjusted based on the user preference.

Diffusion-Stego does not require fine-tuning pre-trained models or training additional models such as extractors or decoders. By using well-learned pre-trained models, Diffusion-Stego generates high-quality stego images, FID score (Heusel et al., 2017) of 2.77 on FFHQ 64×64 (Karras et al., 2019) images while hiding 1.0 bits per pixel (bpp) messages. Additionally, using pre-trained diffusion models trained on AFHQv2 64×64 (Choi et al., 2020), Diffusion-Stego achieves hiding 6.0 bpp messages with high extracted message accuracy. Furthermore, we show that Diffusion-Stego can be easily applied to text-to-image models (Ramesh et al., 2022; Saharia et al., 2022), such as Stable diffusion (Rombach et al., 2022), by leveraging only secret messages and text prompts.

## 2 PRELIMINARIES

### 2.1 GENERATIVE STEGANOGRAPHY

In generative steganography, two players, the sender and the receiver, communicate secret messages $\mathbf{M}$ through generated stego image $\mathbf{X}_S$. Unlike traditional steganography methods (Morkel et al., 2005), the sender in generative steganography uses a generator $G$ to generate $\mathbf{X}_S$ from $\mathbf{M}$ without cover image $\mathbf{X}_C$. The receiver extracts secret messages $\mathbf{M}'$ from $\mathbf{X}_S$ using an extractor $E$. The process is defined as follows :

$$G(\mathbf{M}) = \mathbf{X}_S, \quad E(\mathbf{X_S}) = \mathbf{M}'. \tag{1}$$

In generative steganography, both image quality and extracted message accuracy are significant. Image quality is an indicator of how stego images are photorealistic like real images. The stego images should be visually and statistically similar to the real images to deceive the steganalyzer. Additionally, the generator needs to produce stego images in such a way that the receiver can extract the secret messages. In Diffusion-Stego, we utilize diffusion models as both a generator and an extractor of generative steganography.

### 2.2 DIFFUSION MODELS

Diffusion models (Sohl-Dickstein et al., 2015; Ho et al., 2020) generate images through an iterative denoising process from Gaussian noise. The sampling process can be viewed as solving the ODE process, with time $t$ going from $T$ to 0, using deterministic samplers (Song et al., 2021). This process is termed probability flow ODE (Song et al., 2020) and follows below equation:

$$\mathrm{d}\mathbf{x} = [\mathbf{f}(\mathbf{x}, t) - \frac{1}{2}g(t)^2\mathbf{s}_\theta(\mathbf{x}, t)]\mathrm{d}t, \tag{2}$$

where $\mathbf{f}$ is drift coefficient, $g$ is diffusion coefficient and $\mathbf{s}_\theta$ is score function trained as neural network. The score function estimates $\nabla_{\mathbf{x}}\log p_t(\mathbf{x})$, where $p_t(\mathbf{x})$ is the probability distribution of $\mathbf{x}(t)$.

Through Equation equation 2, diffusion models generate the image $\mathbf{x}(0)$ from Gaussian noise $\mathbf{x}(T) = \sigma_T\mathbf{z}$, where $\sigma_T$ is constant of $T$ and $\mathbf{z}$ is standard Gaussian noise, $\mathbf{z} \sim \mathcal{N}(0, I)$. We

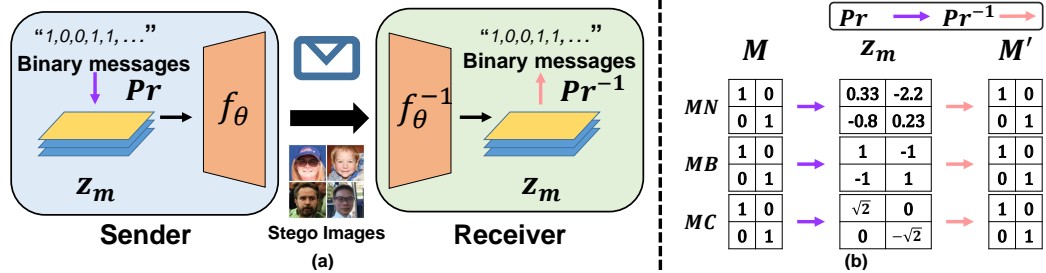

Figure 1: The overview of Diffusion-Stego. (a) Generative steganography process of Diffusion-Stego. (b) The example of three message projection processes when messages are '1001'.

can establish a bijective function between $\mathbf{z}$ and image $\mathbf{x}(0)$ using ODE, namely, $\mathbf{x}(0) = f_\theta(\mathbf{z})$. The function $f_\theta$ is invertible, we can express another equation $\mathbf{z} = f_\theta^{-1}(\mathbf{x}(0))$.

# 3 METHOD

## 3.1 DIFFUSION-STEGO

**Settings** We propose Diffusion-Stego, a new generative steganography framework leveraging pre-trained diffusion models. Diffusion-Stego can hide $n$ bpp binary messages, $\mathbf{M} \in \{0, 1\}^{n \times W \times H}$ in images $\mathbf{X} \in \mathbb{Z}^{3 \times W \times H}$, where $W$ and $H$ denote the width and height of images.

In Diffusion-Stego, we consider two players, the sender and the receiver. The sender projects the secret binary messages $\mathbf{M}$ into Gaussian noise and generates stego images $\mathbf{X}_S$ from the projected noise using a diffusion model. Then, the receiver extracts the hidden noise using the same diffusion model and projects it onto the binary message. We note that the sender and the receiver should share the same diffusion model and the projection process to use Diffusion-Stego. There are no restrictions on the shared diffusion models, so two players can use any pre-trained diffusion models.

Wei et al. (2022b) demonstrated that saving images as float type using TIFF format resulted in higher performance in extracted message accuracy than saving as integer type using PNG or JPEG formats. However, in our research, we mainly use PNG format to save images because integer type formats are more widely used than TIFF. We will present our result in Section 4, including the results of using the TIFF format.

**Procedure** Generative steganography requires two models: $G$, which generates images from messages, and $E$, which extracts messages from images. Previous works have trained two separate models for $G$ and $E$, similar to the $f_\theta$ and $f_\theta^{-1}$ described in Section 2.2. However, both models can be performed by a single diffusion model. Thus, we can use diffusion models as generative steganography, provided that message projection projects $\mathbf{M}$ into the same domain as Gaussian noise $\mathbf{z}$. We use deterministic samplers such as DDIM sampler (Song et al., 2021) or Heun's sampler of EDM (Karras et al., 2022) for the invertible function $f_\theta$.

In Diffusion-Stego, we edit Gaussian noise $\mathbf{z}$ with message noise $\mathbf{z}_m$, where $\mathbf{z}_m$ is the noise hiding $\mathbf{M}$. The number of channels of $\mathbf{z}_m$ to hide messages depends on the message quantity. When the length of messages is $n$ bpp, we use $n$ channels of $\mathbf{z}_m$ while retaining the noise in the unused channels. Messages projection $Pr$ is function that maps $\mathbf{z}$ and $\mathbf{M}$ into $\mathbf{z}_m$, $\mathbf{z}_m = Pr(\mathbf{z}, \mathbf{M})$. If $Pr$ is invertible, we can generate a stego image $\mathbf{X}_S$ and extract hidden messages $\mathbf{M}'$ by solving the ODE process, $\mathbf{x}_s(0) = f_\theta(Pr(\mathbf{z}, \mathbf{M}))$ and $\mathbf{M}' = Pr^{-1}(f_\theta^{-1}(\mathbf{x}_s(0)))$ while considering $\mathbf{X}_S$ as $\mathbf{x}_s(0)$, as shown in Figure 1(a).

## 3.2 CHALLENGING PROBLEMS OF DIFFUSION-STEGO

This section introduces two challenging problems when using diffusion models for generative steganography.

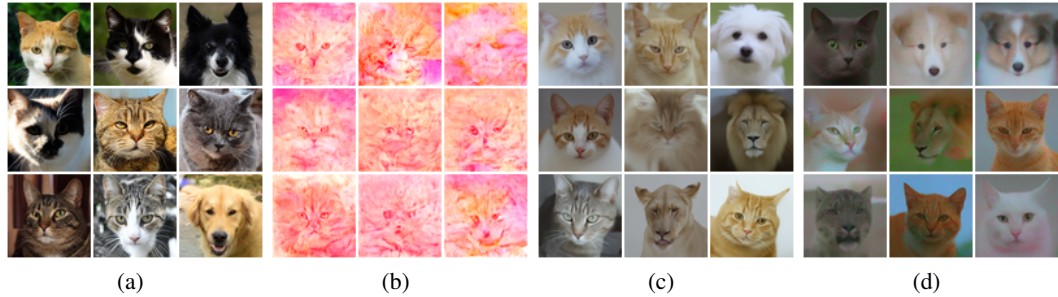

(a)            (b)            (c)            (d)

Figure 3: Images generated by diffusion models. (a) Normally generated images. (b) Collapsed images hiding 1.0 bpp, mean of $\mathbf{z}_m$ differs from $\mathbf{z}$. (c) Collapsed images hiding 1.0 bpp, variance of $\mathbf{z}_m$ differs from $\mathbf{z}$. (d) Collapsed images hiding 1.0 bpp using se-S2IRT (Zhou et al., 2022) algorithm (each value of $\mathbf{z}_m$ is not independent). Additional samples of image collapse are provided in Appendix A.

**Image collapse** In Diffusion-Stego, we utilize the invertible property of the deterministic sampler of diffusion models. Diffusion models have learned to map Gaussian noise to images. If the projection $Pr$ is defined in a simplistic manner, the distribution of $\mathbf{z}_m$ can differ from that of Gaussian noise. In this case, image quality may be harmed or even collapse, as shown in Figure 3. We refer to this issue as *image collapse*.

To prevent the image collapse, we need to make the distribution of $\mathbf{z}_m$ similar to that of Gaussian noise. We identify three conditions which $\mathbf{z}_m$ need to satisfy: **(1)** The **mean** of $\mathbf{z}_m$ is close to 0: if the mean is higher than 0, the output image becomes white, while if it is lower than 0, it becomes dark. **(2)** The **variance** of $\mathbf{z}_m$ is similar to 1: when the variance is too high or too low, the output image may not be properly denoised or may be oversimplified. **(3)** The values of $\mathbf{z}_m$ are **independent**: when the values are not independent, diffusion models do not work normally. Figure 3 illustrates the results when $\mathbf{z}_m$ does not meet these conditions.

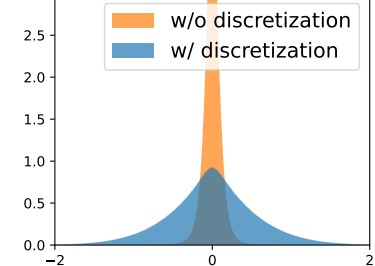

Figure 2: Density distribution of $\mathbf{z} - \mathbf{z}'$, where the error between the input noise $\mathbf{z}$ and the extracted noise $\mathbf{z}' = f_\theta^{-1}(f_\theta(\mathbf{z}))$. The orange line: the error without discretization. The blue line: the error with discretization into integers.

**Extraction error** As the deterministic sampler solves the ODE process with a numerical integrator, errors accumulate in both the forward and backward processes of the ODE. This can lead to a decrease in the extracted message accuracy, which we term as *extraction error*. Additionally, in the steganography process, the sender needs to save images in integer formats, such as PNG or JPEG, to send to the receiver. Discretization during image saving amplifies the error. In Figure 2, we show the accumulated error between input noise and extracted noise. Small errors occur during the numerical integrator (the orange line), and the errors become larger due to discretization (the blue line).

### 3.3 MESSAGE PROJECTION

In this section, we describe message projection, the key ingredient for solving both problems, the image collapse and the extraction error. We suggest three options for message projection, which are designed depending on which problem to focus on. Examples of our message projections are shown in Figure 1(b), and the corresponding mathematical expressions are exhibited in Table 1

Table 1: Mathematical expression of message noise. $M$ denotes messages, $n$ represents random noise, $s$ is randomly sampled sign value from the set $\{-1, 1\}$.

| Options | $z_m$ |
|---------|-------|
| MN | $abs(n) \cdot (2M - 1)$ |
| MB | $2M - 1$ |
| MC | $\sqrt{2}M \cdot s$ |

**Message to Noise (MN)** We propose MN projection $Pr_N$ to solve the image collapse. The projection $Pr_N$ maps the messages to $\mathbf{z} \sim \mathcal{N}(0, I)$, so that the distribution of message noise $\mathbf{z}_m$ is equivalent to that of Gaussian noise.

To implement the MN projection, we first sample standard Gaussian noise $\mathbf{z}$. Then, we project $\mathbf{z}$ into $\mathbf{z}_m$ using the following rule: change the sign of $\mathbf{z}$ to a positive number where $\mathbf{M}$ is 1, and to a negative number where $\mathbf{M}$ is 0. The receiver can extract the messages by applying the inverse projection $Pr_N^{-1}$, which checks whether the value of extracted $\mathbf{z}_m$ is greater than 0 or not.

Since the probability distribution of $\mathbf{z}_m$ is the same as that of the Gaussian noise, generated images $f_\theta(Pr_N(\mathbf{z}, \mathbf{M}))$ are challenging to distinguish from normally generated images. However, using MN projection is vulnerable to the extraction error because some values of $\mathbf{z}_m$ are situated near the decision boundary of $Pr_N^{-1}$, which is 0. Therefore, we suggest other projection options to improve the extracted message accuracy.

**Message to Binary (MB)** The MB projection is designed to solve the aforementioned problem, the extraction error. In the MB projection, the receiver identifies the messages by inverse projection $Pr_B^{-1}$, the same as that of the MN projection. $Pr_N^{-1}$. The MB projection $Pr_B$ maps the values of $\mathbf{z}_m$ as far as possible from 0, which corresponds to the decision boundary of $Pr_B^{-1}$.

To maximize the minimum distance from the boundary, $Pr_B$ equals all the values that denote the same messages. To ensure that the variance of $\mathbf{z}_m$ becomes 1, $Pr_B$ set the value of $\mathbf{z}_m$ to 1 where $\mathbf{M}$ is 1 and to -1 where $\mathbf{M}$ is 0.

**Message to Centered Binary (MC)** While the MN projection resolves the image collapse, it performs worse in terms of extracted message accuracy than the MB projection. The MB projection well addresses the extraction error. However, the distribution of $\mathbf{z}_m$ deviates from that of Gaussian noise, which causes a slight degradation in image quality. Therefore, we suggest a compromise between the two projections, called the MC projection.

Similar to the MB projection, MC projection $Pr_C$ projects the values which denote the same messages into coherent values. Since the mode of Gaussian noise is 0, we set the value of $\mathbf{z}_m$ to 0, where $\mathbf{M}$ is 0. When $\mathbf{M}$ is 1, we randomly map the value of $\mathbf{z}_m$ to either $\sqrt{2}$ or $-\sqrt{2}$, so that the mean and variance of $\mathbf{z}_m$ are equal to those of Gaussian noise. Inverse projection $Pr_C^{-1}$ checks the values of $\mathbf{z}_m$ are close to $\sqrt{2}, -\sqrt{2}$, or 0.

The MC projection performs higher extracted message accuracy than the MN projection and better sample quality than the MB projection, which will be demonstrated in Section 4.

### 3.4 TRICK OF HIDING LARGE MESSAGES

Generally, input noise for diffusion models consists of 3 channels. When applying the projections referred to in Section 3.3, the maximum capacity of secret messages is 3.0 bpp. In order to conceal more messages than 3.0 bpp, we should hide more than 1.0 bpp messages in a single channel.

We hide multiple bits following MB, which we call Multi-bits projection. Two bits messages consist of four cases: 00, 01, 10, and 11. We set four values and keep them as far away from each other as possible while maintaining the mean and variance of $\mathbf{z}_m$. For 2 bits, we define the value as $-3/\sqrt{5}, -1/\sqrt{5}, 3/\sqrt{5}$, or $1\sqrt{5}$, where $\mathbf{M}$ is 00, 01, 10, or 11. We can hide 6.0 bpp messages by hiding 2 bits in each channel and more messages by applying this projection.

### 4 EXPERIMENTS

**Datasets and pre-trained models** We consider three commonly used image datasets for generative models: CIFAR-10 (Krizhevsky et al., 2009), FFHQ 64×64 (Karras et al., 2019) and AFHQv2 64×64 (Choi et al., 2020). Several previous works (Song et al., 2020; Karras et al., 2022; Xu et al., 2022; 2023) provide pre-trained models trained on these datasets. In our experiments, we use pre-trained models and deterministic Heun's sampler of EDM (Karras et al., 2022). For the FFHQ 64×64 and AFHQv2 64×64 datasets, we processed $f_\theta$ and $f_\theta^{-1}$ with 40 inference steps, while for the CIFAR-10 dataset, we used 18 inference steps. We conduct our experiments using 4 Nvidia Titan Xp GPUs.

Our method can utilize high-resolution pre-trained models, as demonstrated in the Appendices E and F. We show that our model can be applied to 256x256 pre-trained models and Stable Diffusion.

Table 2: Comparison of extracted message accuracy (Acc, %), anti-detection ability (Pe), and image quality (FID) with baseline methods. [†]: our re-implementation.

| Method | 1.0 bpp | | | 2.0 bpp | | | 3.0 bpp | | |
|---|---|---|---|---|---|---|---|---|---|
| | Acc $\uparrow$ | Pe $\uparrow$ | FID $\downarrow$ | Acc $\uparrow$ | Pe $\uparrow$ | FID $\downarrow$ | Acc $\uparrow$ | Pe $\uparrow$ | FID $\downarrow$ |
| GSN[†] | 97.15 | 0.183 | 13.4 | 79.62 | 0.022 | 24.8 | 72.74 | 0.049 | 30.4 |
| S2IRT[†] | **99.94** | 0.003 | 72.6 | 97.79 | 0.002 | 78.2 | 97.13 | 0.003 | 67.8 |
| Diffusion-Stego | 98.12 | **0.427** | **2.77** | **98.19** | **0.361** | **3.30** | **98.76** | **0.310** | **4.30** |

**Metrics**   We evaluate extracted message accuracy, anti-detection ability, and image quality of our methods. The accuracy (Acc) measures the accuracy of the extracted message, which may be distorted through the steganography process $f_\theta$ and $f_\theta^{-1}$. We calculate Acc as follows: Acc $= 1 - \frac{\mathbf{M} \oplus \mathbf{M}'}{\text{len}(\mathbf{M})}$, where $\mathbf{M}$ is original binary messages, $\mathbf{M}'$ is extracted binary messages through the steganography process, $\oplus$ is XOR operator, and $\text{len}(\mathbf{M})$ is length of the messages. The detection error (Pe) is an indicator of the performance of classifier models. Pe is defined as follows: Pe $= \min_{P_{FA}} \frac{1}{2}(P_{FA} + P_{MD})$, where $P_{FA}$ and $P_{MD}$ are the rates of false-alarm and miss-detection errors. A Pe value of 0.5 means that the classifier can not distinguish two classes completely. We use Xu-Net (Xu et al., 2016) models to evaluate Pe of steganalyzer models and assess the anti-detection ability of our method. Frechet inception distance (FID) score (Heusel et al., 2017) is an image quality assessment indicator, where a lower FID score indicates better image quality. Bit per pixel (bpp) is a unit for the message quantity hiding in images. We calculate bpp as follows: $\frac{\text{len}(\mathbf{M})}{W \times H}$,where $W$ and $H$ are width and height of image.

In our experiments, We sampled 6,000 stego images from each model to calculate the accuracy. We divide the stego images into 5,000 training sets and 1,000 test sets for training and evaluating steganalyzer models. We train Xu-Net on 5,000 stego images and 5,000 real images and test on 1,000 stego images and 1,000 real images for each steganography model, following in Zhou et al. (2022). We sample 50,000 stego images that hide random messages to calculate FID scores.

**Steganalyzer settings**   As generative steganography models do not have cover images, third-party players cannot train their steganalyzer models. Therefore, we utilize the real images as a substitute of cover images, assuming that third-party players would adopt the strict strategy in their steganalysis.

**Baseline**   We select two baseline generative steganography models which can hide messages above 1.0 bpp, Generative Steganography Network (GSN) (Wei et al., 2022a) and Secret to Image Reversible Transformation (S2IRT) (Zhou et al., 2022).

GSN is a GAN-based (Goodfellow et al., 2020) method that consists of four models: generator, discriminator, steganalyzer, and extractor. In our experiments, we train each GSN model from scratch with different payload settings. S2IRT is a generative steganography method that applies Glow (Kingma & Dhariwal, 2018) models. We train Glow and use Separate Encoder based S2IRT (SE-S2IRT) scheme. The SE-S2IRT scheme splits random values into $K$ clusters and assigns each message to the corresponding cluster based on the order of values. Increasing $K$ leads to a higher message capacity but lower accuracy. In our experiments, we choose a low value of $K$ to optimize extracted message accuracy.

**Discretization**   In our experiments, we discretize stego images into integer values and save them in PNG format. Wei et al. (2022b) proposed that using the TIFF format shows good performance in terms of extracted message accuracy because TIFF format saves an image in continuous values. Following Wei et al. (2022b), we also conduct experiments using TIFF format and present the results in Appendix C.

### 4.1 COMPARISON WITH BASELINE MODELS.

We compare Diffusion-Stego with two high-capacity generative steganography models, GSN and S2IRT, which are trained on the FFHQ 64×64 dataset. The comparison is performed using the MB projection in three payload settings: 1.0 bpp, 2.0 bpp, and 3.0 bpp.

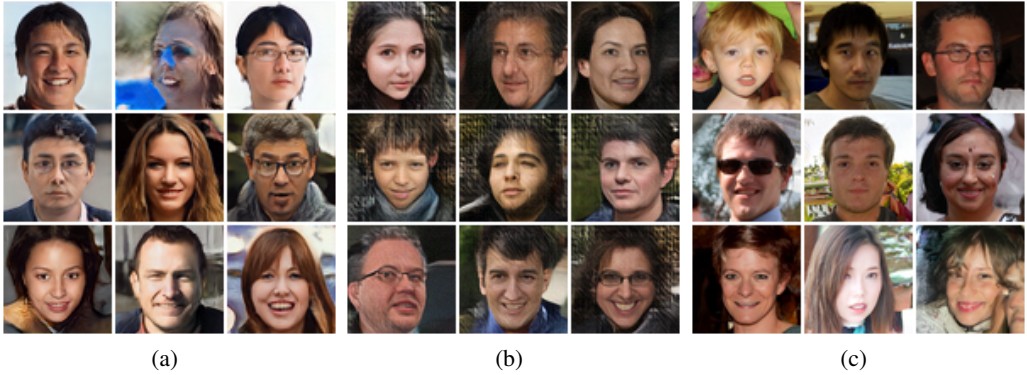

(a)             (b)             (c)

Figure 4: FFHQ 64×64 stego images hiding 1.0 bpp messages. (a) GSN, (b) S2IRT, (c) Diffusion-Stego. Diffusion-Stego is superior to the baseline methods in sample quality.

Table 3: Ablation study results of three projection options on FFHQ 64×64 and AFHQv2 64×64 datasets. The original EDM models have FID scores of 2.39 on FFHQ and 2.17 on AFHQv2.

| Datasets | Projections | 1.0 bpp | | | 2.0 bpp | | | 3.0 bpp | | |
|---|---|---|---|---|---|---|---|---|---|---|
| | | Acc ↑ | Pe ↑ | FID ↓ | Acc ↑ | Pe ↑ | FID ↓ | Acc ↑ | Pe ↑ | FID ↓ |
| FFHQ | MN | 88.00 | 0.422 | **2.41** | 86.75 | **0.433** | **2.42** | 87.06 | **0.427** | **2.45** |
| | MB | **98.12** | 0.427 | 2.77 | **98.19** | 0.361 | 3.30 | **98.76** | 0.310 | 4.30 |
| | MC | 93.17 | **0.445** | 2.58 | 91.97 | 0.414 | 2.75 | 93.09 | 0.409 | 3.11 |
| AFHQv2 | MN | 87.32 | 0.399 | **2.14** | 85.68 | 0.390 | **2.20** | 86.64 | 0.403 | **2.13** |
| | MB | **98.03** | 0.396 | 2.21 | **98.57** | 0.388 | 2.35 | **99.19** | 0.376 | 2.46 |
| | MC | 92.65 | **0.407** | 2.22 | 91.00 | **0.404** | 2.26 | 93.40 | **0.405** | 2.26 |

The results are shown in Table 2. The results presented in Figure 4 and Table 2 show that Diffusion-Stego outperforms the other baseline models in terms of anti-detection ability and image quality. S2IRT shows higher accuracy than the other two models when the message capacity of stego images is 1.0 bpp. However, when the message capacity is higher than 1.0 bpp, Diffusion-Stego showed higher accuracy than S2IRT. Although S2IRT achieves high accuracy (99.94% at 1.0 bpp) when the number of the clusters $K$ is 2, its message capacity is limited to 1.5 bpp. To hide 2.0 bpp messages, $K$ should be at least 3, which decreases the extracted message accuracy. GSN shows competitive accuracy in the payload setting of 1.0 bpp, but it decreases rapidly as the payload increases.

## 4.2 ABLATION STUDY

We compare the performance of our message projection options in our experiments using the FFHQ and AFHQv2 datasets. The results of our comparison are presented in Table 3. Using the MB projection outperforms the other two projections in Acc. When hiding small messages with payloads of 1.0 bpp, the Pe and FID scores of each projection are similar. However, hiding large messages, such as with payloads of 3.0 bpp, the anti-detection ability and image quality of using the MB projection decreases rapidly compared to using the MN projection. In the FFHQ dataset, the MC projection shows compromised results of Acc, Pe, and FID between the MN projection and the MC projection as the payload of messages increases.

## 4.3 PERFORMANCE OF HIDING VARIOUS BPP MESSAGES

We evaluate Diffusion-Stego on various payload settings, ranging from 1.0 bpp to 6.0 bpp for each dataset. In payloads from 1.0 to 3.0 bpp, we use the MB projection. In payloads from 4.0 and 5.0 bpp, we use both the MB projection and the Multi-bits projection. In 6.0 bpp payloads, we only use the Multi-bits projection.

Figure 5 shows the stego images generated by AFHQv2 models. As shown in Table 4, using the MB projection alone (from 1.0 to 3.0 bpp) results in higher extracted message accuracy compared to using the Multi-bits projection. However, using the Multi-bits projection (from 4.0 to 6.0 bpp) pro-

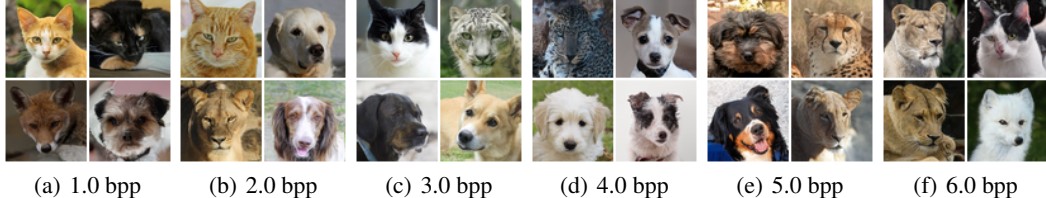

| (a) 1.0 bpp | (b) 2.0 bpp | (c) 3.0 bpp | (d) 4.0 bpp | (e) 5.0 bpp | (f) 6.0 bpp |

Figure 5: AFHQv2 64×64 images which are hiding messages with pre-trained EDM models. Diffusion-Stego can hide 6.0 bpp messages without the image collapse.

Table 4: Results of various message payloads, 1.0 to 6.0 bpp. The original EDM models have FID scores of 2.39 on FFHQ, 2.17 on AFHQv2, and FID scores of 1.97 on CIFAR-10.

| Datasets | metric | 1.0 | 2.0 | 3.0 | 4.0 | 5.0 | 6.0 |
|---|---|---|---|---|---|---|---|
| AFHQv2 64×64 | Acc ↑ | 98.03 | 98.57 | 99.19 | 96.39 | 93.15 | 91.93 |
| | Pe ↑ | 0.396 | 0.388 | 0.376 | 0.364 | 0.390 | 0.394 |
| | FID ↓ | 2.21 | 2.35 | 2.46 | 2.42 | 2.35 | 2.34 |
| FFHQ 64×64 | Acc ↑ | 98.12 | 98.19 | 98.76 | 95.57 | 92.38 | 91.12 |
| | Pe ↑ | 0.427 | 0.361 | 0.310 | 0.334 | 0.345 | 0.385 |
| | FID ↓ | 2.77 | 3.30 | 4.30 | 3.90 | 3.67 | 3.37 |
| CIFAR-10 | Acc ↑ | 95.38 | 95.07 | 95.19 | 89.93 | 86.43 | 84.83 |
| | Pe ↑ | 0.434 | 0.460 | 0.434 | 0.417 | 0.446 | 0.441 |
| | FID ↓ | 2.09 | 2.30 | 2.66 | 2.48 | 2.37 | 2.31 |

vides better anti-detection ability and image quality. This is because the distribution of $\mathbf{z}_m$ projected by the Multi-bits projection is more similar to that of Gaussian noise than that of the MB projection. When the model is trained on the CIFAR-10 dataset, the extracted message accuracy is lower than those trained on other datasets. This is due to the susceptibility of generated CIFAR-10 images to discretization, which will be presented in Appendix C.

As the number of bits to hide in channels increases, the extracted message accuracy decreases due to the same reason as the MN projection. We hide 9.0 bpp messages using EDM models trained on the AFHQv2 dataset and the Multi-bits projection. The extracted message accuracy from the stego images hiding 9.0 bpp messages is 83.18%.

### 4.4 COMPARISON WITH IMAGE STEGANOGRAPHY

We compare Diffusion-stego with image steganography, FNNS (Kishore et al., 2021). Specifically, we employ the CIFAR-10 pfgm++ (Xu et al., 2023) model and hiding 3 bpp messages using the MB method. We integrated diffusion models as the decoder within FNNS.

The results of this comparison are presented in Table 5. As the number of iterations in the FNNS task increases, the accuracy of the message increases, but the anti-detection ability decreases. Image steganography methods have an advantage in terms of message accuracy, while generative steganography methods excel in anti-detection ability. We provide the comparison results with LISO (Chen et al., 2022) in the Appendix G.

## 5 RELATED WORK

### 5.1 STEGANOGRAPHY WITH DEEP LEARNING

Steganography methods employing deep learning techniques have emerged, encompassing various machine learning tasks. Some approaches (Baluja, 2017; Zhang et al., 2019; Zhu et al., 2018; Jing et al., 2021) utilize encoder-decoder architectures to conceal secrets within provided cover images using trained models. Other methods (Kishore et al., 2021; Chen et al., 2022) employ optimization techniques to manipulate the cover images in such a way that the secret becomes visible when decoding the altered cover images.

Table 5: Comparison result between Diffusion-stego and FNNS using same diffusion models. The result of 0 iterations is that of Diffusion stego.

| Metric | 0 iters (Ours - MB ) | 1 iter | 2 iters | 3 iters | 4 iters |
|--------|----------------------|--------|---------|---------|---------|
| Acc ↑ | 95.60 | 99,47 | 99.67 | 99.69 | **99.70** |
| Pe ↑ | **0.456** | 0.435 | 0.218 | 0.211 | 0.179 |
| FID ↓ | **2.51** | 2.66 | 3.05 | 3.11 | 3.19 |

## 5.2 GENERATIVE STEGANOGRAPHY

Generative steganography is a method where generative models synthesize images from secret messages without using any cover images. Generative steganography offers several advantages over traditional steganography methods that use cover images. One of the main benefits is that it can avoid detection by steganalysis methods because it does not modify images. Further, steganalysis methods that are trained on such images cannot detect the presence of hidden data.

Early studies of generative steganography hide messages in simple images, such as texture or fingerprint images. Wu & Wang (2014) and Xu et al. (2015) proposed approaches to hide secret messages in texture messages. Li & Zhang (2018) proposed the method to use fingerprint images. These approaches generate low-quality and unnatural images, which are prone to be detected by third-parties.

Steganography approaches using generative models have been proposed to make high-quality and natural stego images. Especially, GAN models (Goodfellow et al., 2020) have been used for generative steganography. Liu et al. (2017) and Zhang et al. (2020) hide messages in label embedding of conditional GANs (Mirza & Osindero, 2014; Odena et al., 2017), Hu et al. (2018); Yu et al. (2021); Wang et al. (2018); Wei et al. (2022a) train new extractor models. Zhou et al. (2022); Wei et al. (2022b) proposed an approach to use invertible Flow models (Kingma & Dhariwal, 2018) to enable high capacity of hidden messages.

## 5.3 DIFFUSION MODELS

Diffusion models (Sohl-Dickstein et al., 2015; Ho et al., 2020; Song et al., 2020) generate images through the stochastic iterative process of denoising from Gaussian noise. While this process incurs a high computational cost, it enables the generation of high-quality images. Several studies (Song et al., 2021; 2020) have proposed deterministic sampling methods for diffusion models. These methods aim to remove the stochastic property of diffusion models while sampling images using an invertible process. Song et al. (2021) proposed an implicit sampling by changing the diffusion process to a non-Markov process. Song et al. (2020) proposed probability flow ODE, which considers sampling processes as ODEs. Liu et al. (2022); Zhang & Chen (2022); Lu et al. (2022); Karras et al. (2022) solve probability flow ODE efficiently using high order numerical integrator. In our approach, we utilize Heun's sampler from the EDM (Karras et al., 2022) to take advantage of the invertible property of the deterministic sampler.

## 6 CONCLUSION

We propose Diffusion-Stego, a novel approach to generative steganography using deterministic samplers of diffusion models. We investigate the factors that affect the quality and extracted message accuracy when diffusion models generate stego images. We suggest three options for message projection, which have the trade-off of image quality, anti-detection, and extracted message accuracy. Our approach can hide large messages (more than 1.0 bpp, even 6.0 bpp with an accuracy of 90%) while maintaining the image quality of pre-trained diffusion models.

Our limitations include a trade-off between image quality, anti-detection ability, and extracted message accuracy. Our observations will enable future research to enhance these capabilities. Furthermore, since our approach uses pre-trained diffusion models, it can be extended to other domains such as video (Ho et al., 2022), sound (Kong et al., 2020), and text (Li et al., 2022).

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
