## A  SAMPLE OF IMAGE COLLAPSE

In this section, we present examples where image collapse occurs. Figure 6 illustrates stego images based on the mean of $\mathbf{z}_m$. When the mean is 0, similar to that of Gaussian noise, the diffusion model successfully generates high-quality images. However, when the mean is higher or lower than 0, the images become brighter or darker, respectively. Figure 7 showcases stego images based on the variance of $\mathbf{z}_m$. When the variance is 1, similar to that of Gaussian noise, the diffusion model successfully generates high-quality images, too. However, when the variance is high, the models fail to generate meaningful images, and when the variance is low, the images are oversimplified. Figure 8 demonstrates three cases where the values of $\mathbf{z}_m$ are not independent.

## B  TRICK FOR HIGHER ACCURACY

We introduce an additional projection called multi-channels projection, which enables the hiding of small messages with higher accuracy compared to other projections.

**Settings**    In this approach, two players, the sender and the receiver, are required to share additional information compared to existing projections. Unlike the projections mentioned in the main paper, they need to share a binary codebook, $\mathbf{C} \in \{0, 1\}^{3 \times W \times H}$.

**Multi-channels projection**    To improve accuracy, the sender hides multiple copies of the same message within a single image following the MB projection, and sends it. However, in this scenario, the values of $\mathbf{z}_m$ are not independent, leading to image collapse as shown in Figure 8(a). To address this issue, we utilize a codebook that is independent of the messages, ensuring the independence of $\mathbf{z}_m$ values. Before generating stego images, the sender modifies the sign of $\mathbf{z}_m$ where C is 0. Similarly, the receiver also modifies the sign of the extracted $\mathbf{z}_m$ where C is 0. By checking the messages multiple times, the receiver achieves higher accuracy.

**Experiments**    We use the same experimental settings as described in Section 4. We consider two cases for the multi-channel projection. In the first case, both players use the same codebook, while in the second case, the players change their codebook for each stego image generated. We hide messages with a capacity of 1.0 bpp, and the messages are replicated to three channels.

The results of the multi-channel projection are presented in Table 6. We observe that using the multi-channel projection yields higher accuracy compared to the MB projection when hiding 1.0 bpp messages. However, when the codebook is not changed, the anti-detection ability of the multi-channel projection decreases, likely due to the similarity among images generated with the same codebook.

## C  MESSAGE ACCURACY ACCORDING TO IMAGE FORMAT

In this section, we present the accuracy results based on quantization to the image formats, PNG and TIFF. We evaluate the accuracy for different datasets and all of our projection techniques, as shown in Table 7.

The results show that saving the images in TIFF format yields higher accuracy compared to the PNG and JPEG formats. Saving images in integer-type formats results in data loss and leads to distortion in the ODE solution of the reversed process of diffusion models.

## D  IMPLEMENTATION DETAILS

In our experiments, we use the official code of PFGM++ (Xu et al., 2023) built upon the official code of EDM (Karras et al., 2022). We adopt the same hyper-parameters as EDM and PFGM++. We set $\sigma_{\max} = \sigma_T = 80, \sigma_{\min} = 0.002$, and $\rho = 7$. These hyper-parameters determine the standard deviation of $x(t)$, which represents the noise in the denoising process. When the number of denoising

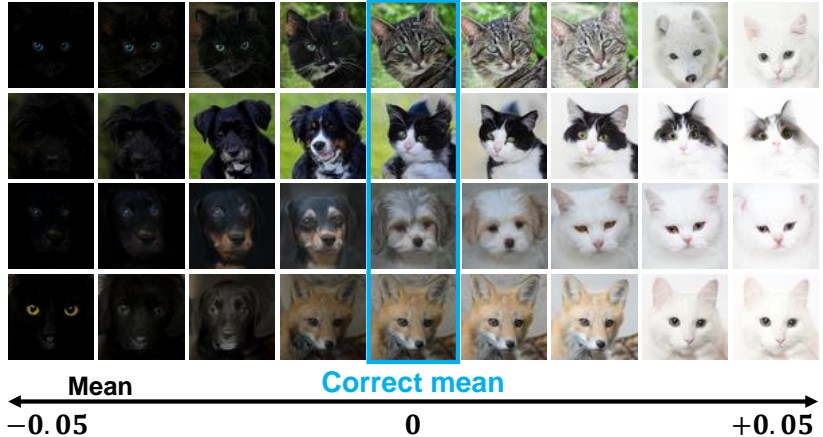

Figure 6: The influence of the mean value on stego images.

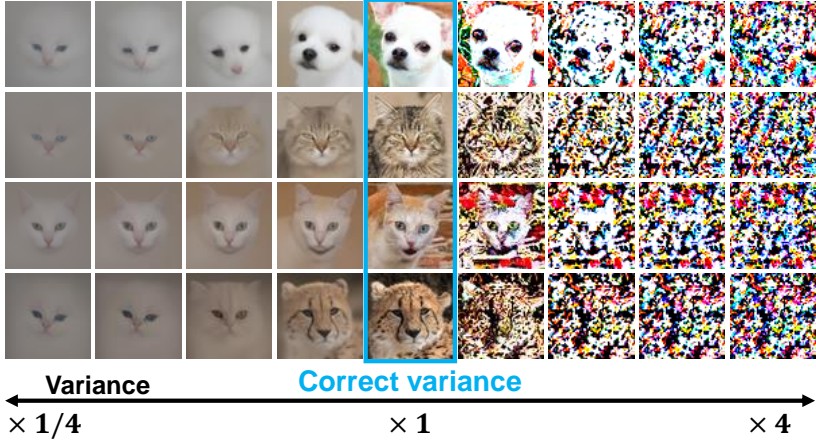

Figure 7: The influence of the variance value on stego images.

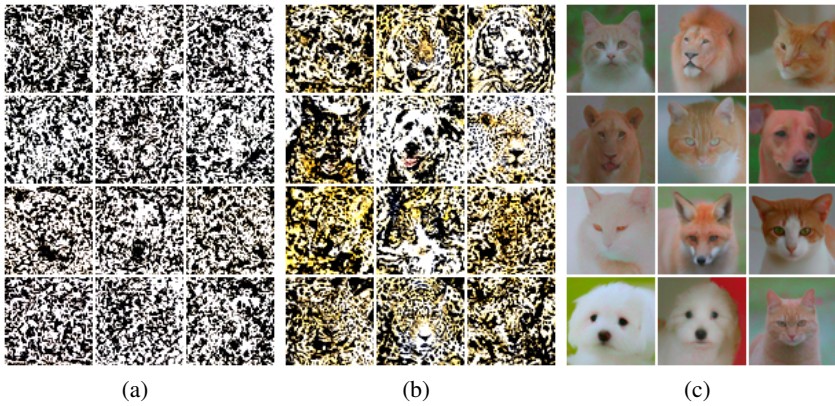

Figure 8: Three examples of stego images where the values of $\mathbf{z}_m$ are not independent. (a) The values within the same pixel share the same sign. (b) The inputs of first and second channel are the same. (c) The algorithm of se-S2IRT with hyper-parameter $K = 2$.

Table 6: Results of the MB projection and multi-channel projection hiding 1.0 bpp messages. *: changing the codebook for each sampling.

| Datasets | Projection | Acc ↑ | Pe ↑ | Fid ↓ |
|---|---|---|---|---|
| FFHQ | MB (1.0 bpp) | 98.12 | 0.427 | 2.77 |
| | Multi-channel | 99.88 | 0.274 | 2.42 |
| | Multi-channel* | 99.81 | 0.307 | 2.46 |
| AFHQ | MB (1.0 bpp) | 98.03 | 0.396 | 2.21 |
| | Multi-channel | 99.76 | 0.321 | 4.43 |
| | Multi-channel* | 99.71 | 0.395 | 4.18 |

inference steps is $N$, the standard deviation of the noise after denoising $i$ times is as follows:

$$\sigma_{\max}^{\frac{1}{\rho}} + \frac{i}{N-1}(\sigma_{\min}^{\frac{1}{\rho}} - \sigma_{\max}^{\frac{1}{\rho}}). \tag{3}$$

For our method, we use Heun's sampler (Karras et al., 2022), which is based on Heun's method (Ascher & Petzold, 1998) for sampling images. Heun's method is a numerical integrator that reduces errors through performing two calculations in each step. In the extraction function, denoted as $f_\theta^{-1}$, we employ Heun's method to estimate $\mathbf{z}_m$, which corresponds to the reversible calculation of Heun's sampler using the same hyper-parameters.

# E    ABLATION STUDIES ON HIGH-RESOLUTION MODELS

We extend our method to high-resolution models, which generate images at a resolution of $256{\times}256$ pixels. We utilize pre-trained EDM models as provided by the authors of the Consistency models (Song et al., 2023) and pre-trained ADM (Dhariwal & Nichol, 2021) models trained on LSUN Cat and Bedroom (Yu et al., 2015) datasets. We generate 100 stego images to evaluate accuracy of message extraction.

**Pre-trained EDM**    The accuracy results of high-resolution pre-trained EDM models are presented in Table 8. While the MB method achieves reasonably favorable performance, it's important to note that the EDM is primarily designed for generating low-resolution images, a resolution of $32{\times}32$ or $64{\times}64$ pixels. Thus, EDM might not be optimal to generate high-resolution images. We anticipate that enhancements in the EDM approach could potentially improve the performance of our method in high-resolution settings. Additional samples images are provided in Figure 9 and 10.

**Pre-trained ADM**    We verify the applicability of the MN method to the pre-trained ADM in these experiments. In this experiment, we reduce the sampling steps to 100 from 1000. Additionally, we skip the final sampling step due to the different structure of ADM compared to EDM. While ADM produces images using discrete time steps, EDM generates images using continuous time steps.

The result of the pre-trained models are shown in Table 9. To stabilize the performance, we incorporate rejection sampling in the experiment. When the accuracy of message extraction is lower than 0.5, we only resample a Gaussian noise and generate stego images while keeping the hidden messages unchanged. We observe that certain stego images generated using the Cat dataset were unsuccessful in recovering the hidden messages, as depicted in Figure 11.

The results of ADM show similar or higher performance in message accuracy compared to the MN method. This can be attributed to the structure of ADM. However, using ADM comes with a high computational cost and results in lower image quality when compared to using pre-trained EDM.

# F    APPLYING ON PRE-TRAINED TEXT-TO-IMAGE MODELS

Our method can be extended to large-scale text-to-image models, which generate images with text guidance. In this case, the sender and the receiver should share two additional pieces of information:

Table 7: The extracted message accuracy from 1000 images. The same set of secrets is applied to all experiments.

| Datasets | Projection | Bpp | w/o quantization | To TIFF | To PNG | To JPEG |
|---|---|---|---|---|---|---|
| AFHQ | MN | 1.0 | 97.71 | 97.62 | 87.35 | 79.25 |
| | | 2.0 | 97.39 | 97.28 | 85.80 | 77.75 |
| | | 3.0 | 97.55 | 97.45 | 86.52 | 78.27 |
| | MB | 1.0 | 100.00 | 100.00 | 98.01 | 90.63 |
| | | 2.0 | 100.00 | 100.00 | 98.18 | 90.09 |
| | | 3.0 | 100.00 | 100.00 | 99.14 | 92.49 |
| | MC | 1.0 | 99.98 | 99.97 | 92.72 | 78.34 |
| | | 2.0 | 99.98 | 99.97 | 90.99 | 75.76 |
| | | 3.0 | 99.99 | 99.98 | 93.48 | 77.93 |
| | Multi-bits | 4.0 | 99.98 | 99.98 | 96.34 | 86.42 |
| | | 5.0 | 99.94 | 99.93 | 93.16 | 81.89 |
| | | 6.0 | 99.91 | 99.89 | 91.58 | 78.94 |
| | | 9.0 | 99.20 | 99.13 | 83.18 | 70.97 |
| FFHQ | MN | 1.0 | 98.60 | 98.45 | 87.92 | 79.23 |
| | | 2.0 | 98.56 | 98.38 | 86.64 | 78.30 |
| | | 3.0 | 98.60 | 98.42 | 87.00 | 78.26 |
| | MB | 1.0 | 100.00 | 100.00 | 97.96 | 89.78 |
| | | 2.0 | 100.00 | 100.00 | 98.10 | 89.75 |
| | | 3.0 | 100.00 | 100.00 | 98.64 | 90.32 |
| | MC | 1.0 | 100.00 | 99.99 | 93.33 | 77.18 |
| | | 2.0 | 100.00 | 99.99 | 92.16 | 76.26 |
| | | 3.0 | 100.00 | 100.00 | 93.46 | 76.85 |
| | Multi-bits | 4.0 | 100.00 | 99.99 | 95.61 | 84.77 |
| | | 5.0 | 99.99 | 99.97 | 92.45 | 80.57 |
| | | 6.0 | 99.99 | 99.97 | 91.08 | 77.82 |
| | | 9.0 | 99.71 | 99.64 | 82.25 | 69.82 |
| CIFAR-10 | MN | 1.0 | 97.55 | 97.38 | 84.51 | 76.40 |
| | | 2.0 | 97.54 | 97.36 | 83.72 | 76.19 |
| | | 3.0 | 97.62 | 97.36 | 84.49 | 76.58 |
| | MB | 1.0 | 100.00 | 100.00 | 94.51 | 85.11 |
| | | 2.0 | 100.00 | 99.99 | 94.23 | 84.22 |
| | | 3.0 | 100.00 | 99.99 | 94.78 | 84.26 |
| | MC | 1.0 | 99.97 | 99.94 | 84.91 | 70.83 |
| | | 2.0 | 99.97 | 99.93 | 84.54 | 69.98 |
| | | 3.0 | 99.98 | 99.95 | 85.26 | 70.65 |
| | Multi-bits | 4.0 | 99.94 | 99.91 | 89.54 | 78.50 |
| | | 5.0 | 99.90 | 99.85 | 86.16 | 75.15 |
| | | 6.0 | 99.86 | 99.79 | 83.98 | 72.51 |
| | | 9.0 | 98.93 | 98.71 | 75.96 | 66.14 |

Table 8: Results of message accuracy using pre-trained high-resolution EDM.

| Method | Cat | | | Bedroom | | |
|---|---|---|---|---|---|---|
| | 1 bpp | 2 bpp | 3 bpp | 1 bpp | 2 bpp | 3 bpp |
| MN | 78.17 | 78.37 | 78.00 | 73.44 | 73.07 | 73.62 |
| MB | **89.3**4 | **90.40** | **90.80** | **82.47** | **82.76** | **83.04** |
| MC | 75.99 | 76.51 | 76.51 | 68.20 | 67.66 | 69.24 |

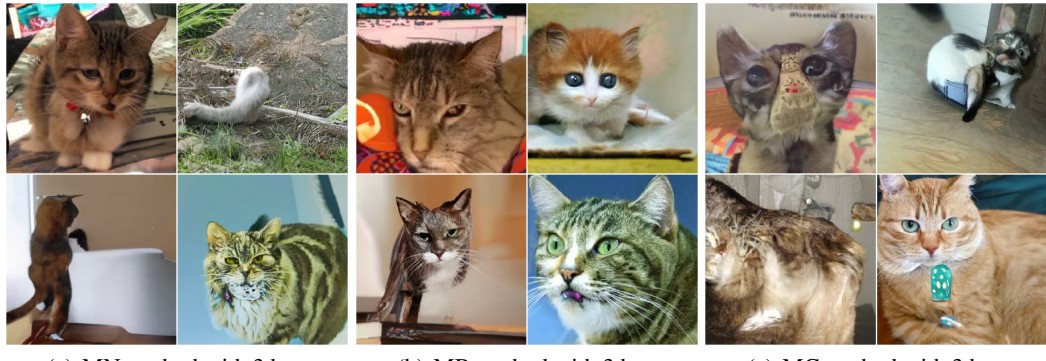

(a) MN method with 3 bpp       (b) MB method with 3 bpp       (c) MC method with 3 bpp

Figure 9: Stego images generated by pre-trained EDM model with LSUN Cat dataset.

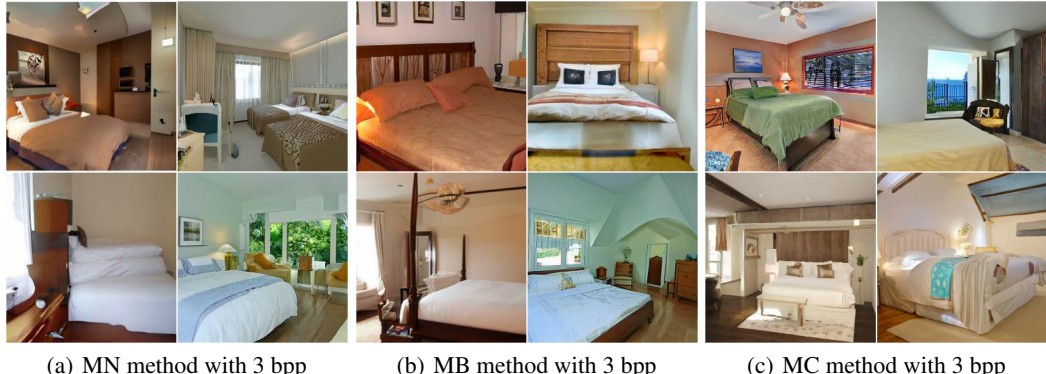

(a) MN method with 3 bpp       (b) MB method with 3 bpp       (c) MC method with 3 bpp

Figure 10: Stego images generated by pre-trained EDM model with LSUN Bedroom dataset.

input text prompt and guidance scale. We use Stable diffusion (Rombach et al., 2022) v1.4, an open-source model trained in the latent space of VAE (Kingma & Welling, 2013). During inference, the diffusion model generates $4\times64\times64$ latent features, which are then decoded to $512\times512$ size images with VAE. Thus, Diffusion-Stego can conceal 0.0625 bpp messages when using Stable diffusion. Figure 12 shows the samples generated by Stable diffusion using the MB projection.

We also measure message accuracy using Stable Diffusion. Specifically, we generated 20 images for each example text provided by the demo page of Stable Diffusion and calculated the accuracy for 0.0625 bpp messages using the MB method. The results for message accuracy are presented in Table 10, and sample images can be found in Figure 13.

The result demonstrate that message accuracy depends on the text conditions, and the task of Diffusion-stego does not affect the generation of text conditions in Stable diffusion.

## G   COMPARISON WITH LISO

We conducted a comparison between Diffusion-stego and LISO (Chen et al., 2022). LISO comprises two tasks: training the encoder and optimization through iterations, similar to FNNS (Kishore et al., 2021). For these experiments, we mainly use FFHQ dataset and pre-trained EDM to train and evaluate both LISO and Diffusion-stego.

**Training LISO**   To compare LISO and Diffusion-stego, we generate FFHQ $64\times64$ images and use them as cover images in LISO model. However, we observed that training LISO model with generated images shows low performance, it is because that LISO is designed to use with high-resolution images. We train LISO model with 1000 FFHQ $1024\times1024$ images, and the ablation

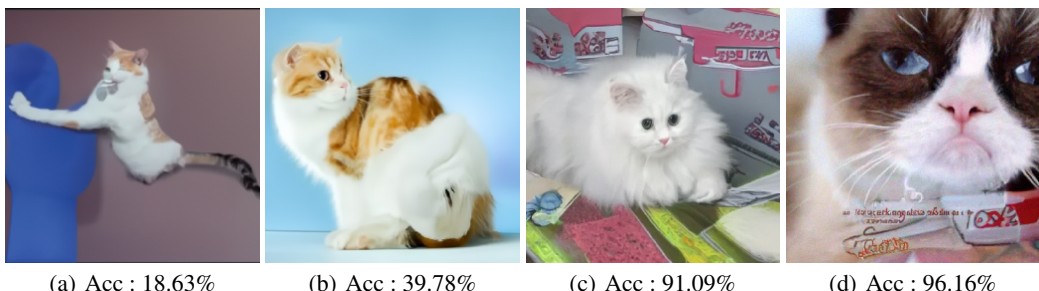

| (a) Acc : 18.63% | (b) Acc : 39.78% | (c) Acc : 91.09% | (d) Acc : 96.16% |

Figure 11: Failure case (a),(b) and successful case (c), (d) of ADM with LSUN Cat dataset.

Table 9: Accuracy results of the pre-trained ADM. If the accuracy is below 0.5, we perform a re-sampling procedure involving only random noise while keeping the message fixed in the rejection sampling task.

| Method | Cat | Cat (Rejection) | Bedroom |
|---|---|---|---|
| MN 1 bpp | 70.89 | 88.74 | 87.20 |
| MN 2 bpp | 73.22 | 88.91 | 87.81 |
| MN 3 bpp | 69.28 | 88.92 | 87.88 |

results are presented in Table I. In comparison experiment, we employ LISO models trained with high-resolution images for evaluation.

**Experiments** In this experiment, we generate the same number of images used to the experiments of the Section 4. We hide 3 bpp message using LISO and Diffusion-stego with the MB method, and Table 12 shows the results. Using the LISO model yielded higher accuracy compared to using Diffusion-stego; images generated with LISO are more distinguishable than those generated with Diffusion-stego.

This result shows that generative steganography methods are effective in concealing secret messages within low-resolution settings. When working with low-resolution images, the objects within the images are often compact. This indicates that there is limited space available for hiding secrets. We hypothesize that the difficulty in training LISO using low-resolution generated images stems from this issue. In such compact conditions, generative steganography methods that can utilize image objects as the secrets, offer advantages for effective secret hiding than image steganography methods.

## H  POTENTIAL SOCIAL IMPACT

We propose a powerful generative steganography that can enhance information security. In scenarios involving wiretapping and vulnerable communications, Diffusion-Stego provides protection against third-party players attempting to access user information. However, it also introduces certain risks as malicious individuals or industrial spies may exploit it to compromise corporate confidentiality. Since our generative steganography approach relies on pre-trained diffusion models, it becomes susceptible to such exploitation. Further research in steganalysis and developing safeguards against the misuse of pre-trained diffusion models is necessary to address these risks.

## I  ADDITIONAL SAMPLES

We present additional samples of stego images generated by pre-trained models trained on AFHQv2 and FFHQ 64×64 datasets in Figure 15, 16, and 17. The stego images generated using the CIFAR-10 dataset are displayed in Figure 18 and 19.

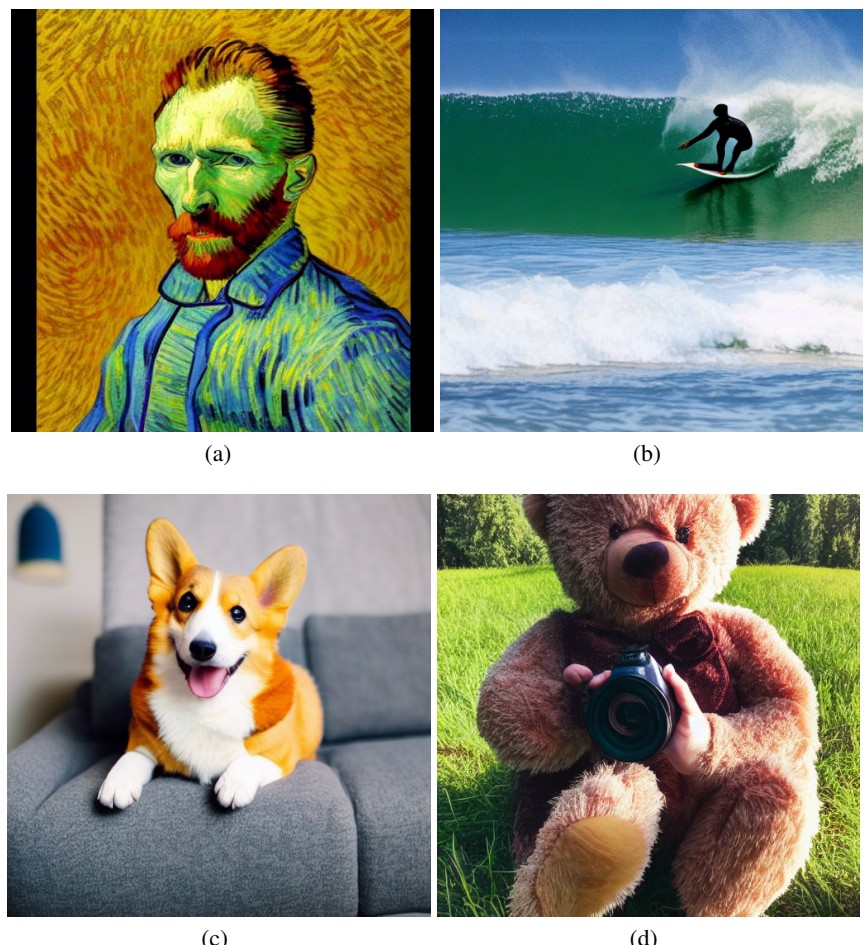

Figure 12: Sample stego images generated by Stable diffusion. Stable diffusion can generate high-quality images while hiding secret messages. (a) 'A painting of Gogh' (Acc: 92.09%). (b) 'A photograph of a surfer' (Acc: 98.25%). (c) 'A photograph of a corgi sitting on a couch' (Acc: 97.04%). (d) 'A photograph of a teddy bear taking a photo' (Acc: 98.04%).

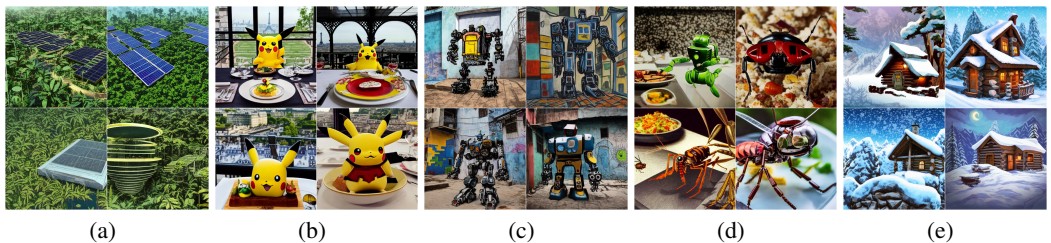

Figure 13: Stego images generated by Stable Diffusion with 5 different text conditions : (a)-A high tech solarpunk utopia in the Amazon rainforest, (b)-A pikachu fine dining with a view to the Eiffel Tower, (c)-A mecha robot in a favela in expressionist style, (d)-an insect robot preparing a delicious meal, (e)-A small cabin on top of a snowy mountain in the style of Disney, artstation.

Table 10: Result of message accuracy using Stable diffusion models and the MB method.

| Metric | Amazon | Eiffel Tower | Mecha robot | Insect robot | Cabin |
|--------|--------|--------------|-------------|--------------|-------|
| Acc↑   | 96.81  | 93.22        | 98.14       | 96.96        | 97.59 |

Table 11: Result of training LISO models with different images. Using high-resolution images for training shows higher performance.

| Models | Acc ↑ | SSIM↑ | PSNR↑ |
|---|---|---|---|
| LISO with generated images ( 64×64 ) | 92.19 | 0.827 | 23.31 |
| LISO with high-resolution images ( 1024×1024 ) | **99.98** | **0.919** | **28.52** |

Table 12: Comparison result of LISO and Diffusion-stego.

| Method | Acc ↑ | Pe ↑ | Fid ↓ |
|---|---|---|---|
| Diffusion-stego (MB) | 98.76 | **0.310** | **4.30** |
| LISO | **99.98** | 0.001 | 74.3 |

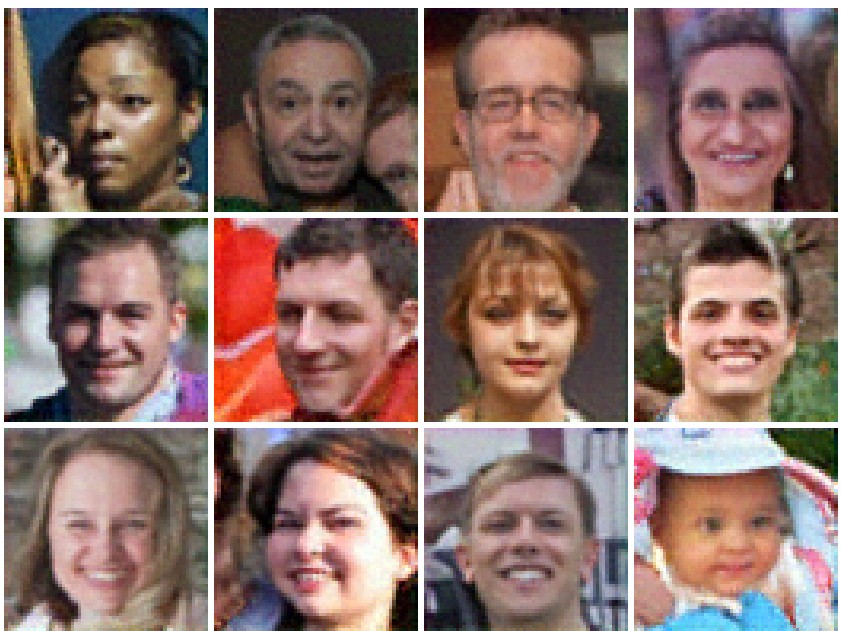

Figure 14: Stego images generated by LISO and pre-trained EDM.

$Pr_N$

$Pr_B$

$Pr_C$

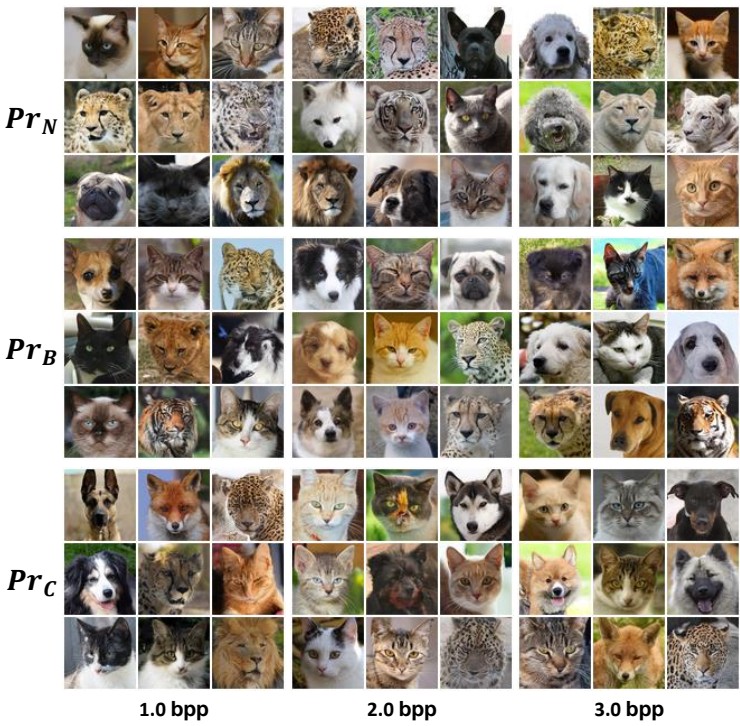

1.0 bpp          2.0 bpp          3.0 bpp

Figure 15: AFHQv2 64×64 stego images with pre-trained EDM models.

$Pr_N$

$Pr_B$

$Pr_C$

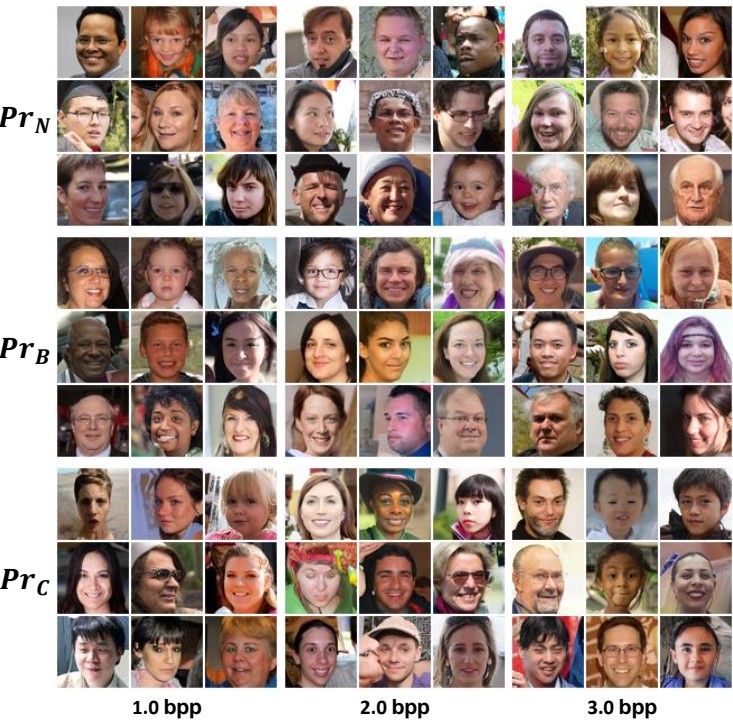

1.0 bpp          2.0 bpp          3.0 bpp

Figure 16: FFHQ 64×64 stego images with pre-trained EDM models.

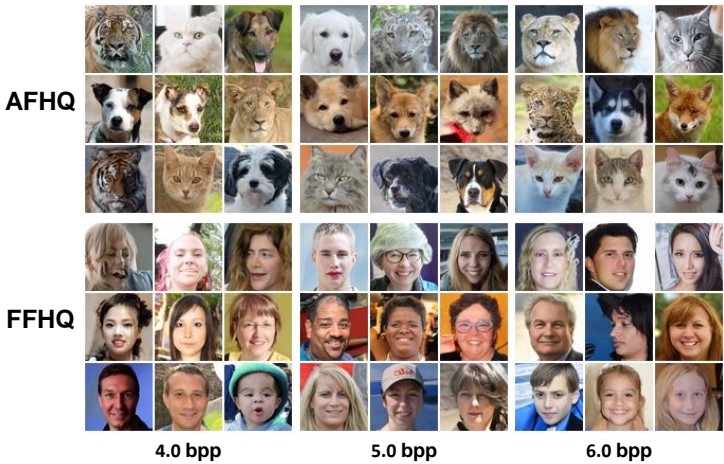

Figure 17: Stego images with pre-trained EDM models and the multi-bits projection.

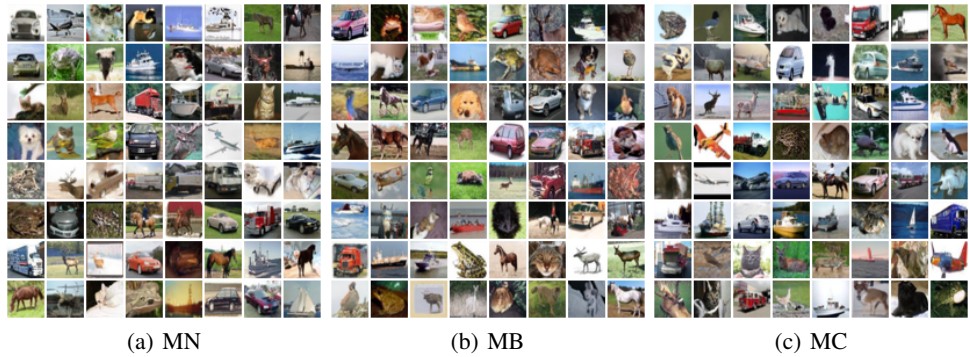

(a) MN            (b) MB            (c) MC

Figure 18: CIFAR-10 stego images with pre-trained EDM models hiding 3.0 bpp messages.

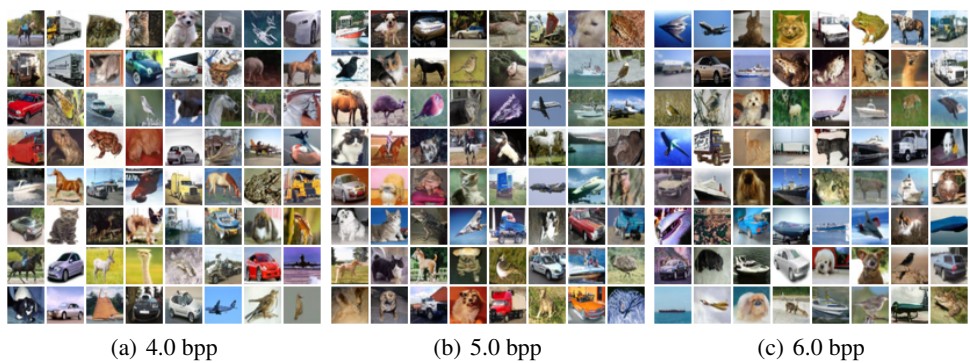

(a) 4.0 bpp        (b) 5.0 bpp        (c) 6.0 bpp

Figure 19: CIFAR-10 stego images with pre-trained EDM models and the multi-bits projection.