# OpenReview forum: "Diffusion-Stego: Training-free Diffusion Generative Steganography via Message Projection"
_ICLR.cc/2024/Conference — ICLR 2024 Conference Withdrawn Submission_

### Official Review · Reviewer_xvqX · 2023-10-29

**Soundness:** 3 good
**Presentation:** 2 fair
**Contribution:** 2 fair
**Rating:** 5
**Confidence:** 4

**Summary:**

This paper introduces Diffusion-Stego, a Generative Steganography scheme. It employs ODE-solvers to handle diffusion images/noises, which can be approximately considered as invertible data pairs.

**Strengths:**

The proposed method achieves a high capacity while preserving impressive visual quality.

**Weaknesses:**

1) Apart from MN, both MB and MC alter the original Gaussian distribution, which results in a reduction of diversity in the produced stego images. For instance, the stego samples depicted in Figure 9 and Figure 11 of the supplement appear visually inadequate. In contrast, there are several demonstrably secure generative steganography techniques [A-D] that are able to maintain the original distribution.
2) Slight distortions, as indicated in supplementary materials like image storage quantization, can result in significant performance degradation. Moreover, it's relatively easy to distinguish between AIGC-generated images and natural images [E-F]. Thus, in terms of Security and Robustness, Diffusion-Stego might be less practical in real-world applications.
3) The three mapping rules are hand-crafted, which is too simple. I think using MLPs or some lightweight networks to learn mapping rules may be a better choice.

Ref:

[A] Cachin C. An information-theoretic model for steganography[C]//Information Hiding: Second International Workshop, IH’98 Portland, Oregon, USA, April 14–17, 1998 Proceedings. Berlin, Heidelberg: Springer Berlin Heidelberg, 1998: 306-318.

[B] Yang K, Chen K, Zhang W, et al. Provably secure generative steganography based on autoregressive model[C]//International Workshop on Digital Watermarking. Cham: Springer International Publishing, 2018: 55-68.

[C] Ding J, Chen K, Wang Y, et al. Discop: Provably Secure Steganography in Practice Based on “Distribution Copies”[C]//2023 IEEE Symposium on Security and Privacy (SP). IEEE Computer Society, 2023: 2238-2255.

[D] Kaptchuk G, Jois T M, Green M, et al. Meteor: Cryptographically secure steganography for realistic distributions[C]//Proceedings of the 2021 ACM SIGSAC Conference on Computer and Communications Security. 2021: 1529-1548.

[E] Wang S Y, Wang O, Zhang R, et al. CNN-generated images are surprisingly easy to spot... for now[C]//Proceedings of the IEEE/CVF conference on computer vision and pattern recognition. 2020: 8695-8704.

[F] Lorenz P, Durall R L, Keuper J. Detecting Images Generated by Deep Diffusion Models using their Local Intrinsic Dimensionality[C]//Proceedings of the IEEE/CVF International Conference on Computer Vision. 2023: 448-459.

**Questions:**

The paper writing is good. However, I am not convinced that an ICLR paper is expected to be only incremental in application compared to previous similar works, which also wish to leave real-world situation in future works. It might be the case for some already published paper that robustness is not considered, but I think the stego society should be more aware of it.

Therefore I maintain my initial rating.

---

> ### Author Response · Authors · 2023-11-15
>
> Thank you for your comments about our paper.
>
> # Response to Weakness 1.
>
> Figure 9-(c) and 11 depict images generated using the MN method, which does not distort the distribution. This suggests that the perceived inadequacy in these images may stem from the dataset or the trained model. We consider the quality of the proposed images in our appendix (Figure 9, and 11) is comparable to those proposed samples from [1]. The model trained on LSUN Bedroom images, has demonstrated high quality, suggesting that generating images from the LSUN Cat dataset might pose a challenge.
> Indeed, images generated by the EDM model trained on the  easily learnable afhq-v2 dataset exhibit high quality.
>
> # Response to Weakness 2.
>
> We acknowledge that the mentioned issues raised may be inherent in the broader field of generative steganography research. Nonetheless, our approach, utilizing models specifically trained for image generation, yields higher-quality images in comparison to traditional generative steganography methods. This enhanced image quality contributes to increased indistinguishability from real images.
>
> Moreover, when employing distortion-free methods such as the MN method, steganalyzers are unable to discern whether the generated images are stego images or not.
>
> # Response to Weakness 3.
>
> In our methodology, two players only need to share the same generative model and the method of message projection.
> While it is possible to train new networks for message mapping, this approach significantly increases the amount of information that needs to be shared between the two participants.
> It also imposes an additional burden of learning new models, aside from diffusion models.
> Specifically, if we train an additional network to improve the trade-off between accuracy and image quality, we would need to evaluate the diffusion models during the training process, incurring computational cost.
> Due to these considerations, we propose our training-free message projection, which is free from information-sharing requirements.
>
> [1] Song, Yang, et al. "Consistency models." International Conference on Machine Learning. PMLR, 2023.

---

> ### Comment · Reviewer_xvqX · 2023-11-22
>
> After reading the rebuttal, I think the author's comments are similar to what is written in the paper, with limited new points conveyed. The response to Weakness 3 is relatively good, however the responses to Weakness 1-2 are not. These two issues would affect the practical application value of the proposed method. Especially for Weakness 2, the author agreed that minor distortion can also cause information extraction to fail. However, in my opinion, this problem should not be literally circumvented, but should be solved. If we continue to encourage bypassing the robustness problems in generative steganography, this would only encourage the field to become more useless in the future. In addition, there have been many diffusion-based steganography works recently (I list them down below). Compared to these works, this paper has basically no outstanding features. Therefore, I suggest that the quality of the current version is not up to the overall level of ICLR.
>
> [A] Wei, Ping, Qing Zhou, Zichi Wang, Zhenxing Qian, Xinpeng Zhang, and Sheng Li. "Generative Steganography Diffusion." arXiv preprint arXiv:2305.03472 (2023).
>
> [B] Peng, Yinyin, Donghui Hu, Yaofei Wang, Kejiang Chen, Gang Pei, and Weiming Zhang. "StegaDDPM: Generative Image Steganography based on Denoising Diffusion Probabilistic Model." In Proceedings of the 31st ACM International Conference on Multimedia, pp. 7143-7151. 2023.
>
> [C] Liu, Tengjun, Ying Chen, and Wanxuan Gu. "Deniable Diffusion Generative Steganography." In 2023 IEEE International Conference on Multimedia and Expo (ICME), pp. 67-71. IEEE, 2023.
>
> [D] Yu, Jiwen, Xuanyu Zhang, Youmin Xu, and Jian Zhang. "CRoSS: Diffusion Model Makes Controllable, Robust and Secure Image Steganography.", NeurIPS 2023.

---

> > ### Author Response · Authors · 2023-11-23
> >
> > We appreciate your response to our rebuttal.
> >
> > # Response to Weakness 2.
> > We misunderstood that the performance degradation mentioned by the reviewer, specifically in the context of quantization, was related to the decline in anti-detection ability due to message projection.
> >
> > In the experimental results, when images were saved in JPEG format, our method exhibited a degradation of message accuracy. However, it's important to note that this degradation is attributed not to quantization itself, but rather to the compression method employed by the JPEG format.
> > Existing methods face similar issues with the JPEG format, and our future work aims to address this problem. If there are other works that demonstrate minimal accuracy reduction when utilizing the JPEG format, they could provide valuable insights for improving our method.
> >
> > # Comparison to other works
> >
> > We wish to underscore the distinct advantages of our paper compared to other methodologies suggested by the reviewer. Our approach is characterized by three key features: 1. It leverages pre-trained models, 2. It integrates an EDM-based model and an ODE solver, and 3. It minimizes the volume of information exchanged between the two parties.
> >
> > In contrast to alternative methodologies, in the case of [A], the necessity to train a new model for steganography generation and the introduction of noise during the learning process lead to a degradation in image quality.
> >
> > As for [B], they employ a DDPM model, demonstrating high accuracy and resistance to distortion. However, the adoption of SDE solver (DDPM) introduces significant drawbacks compared to our approach. Firstly, in [B], both players must share as many random noises as the required steps for DDPM sampling, and since the default number of DDPM sampling steps is 1000, both players need to share 999 random noises. Moreover, due to the utilization of an SDE solver, they incur a higher computational cost for image generation compared to ODE-based diffusion models, as employed in our method. Our EDM-based approach requires 35 NFEs to generate CIFAR-10 images, whereas DDPM demands more NFEs. Reports of DDIM [1] indicate that using a small number of steps for SDE solver in DDPM leads to a significant drop in image quality. While it's possible to apply the ODE solver proposed in DDIM to the DDPM model, doing so would render [B]'s proposed methodology unusable.
> > Additionally, the high-quality image generation of models like EDM is compromised when applying [B]'s method due to vulnerabilities in quantization.
> >
> > For [C], training a new model and additional model training (steganalyzer, discriminator, etc.) are required. Furthermore, the stego images generated by their methodology have a lower capacity and significantly lower image quality compared to our approach (FID score of 3.75 for CIFAR-10).
> > [1] Song, Jiaming, Chenlin Meng, and Stefano Ermon. "Denoising diffusion implicit models." arXiv preprint arXiv:2010.02502 (2020).

---

### Official Review · Reviewer_LtLp · 2023-10-30

**Soundness:** 3 good
**Presentation:** 3 good
**Contribution:** 2 fair
**Rating:** 3
**Confidence:** 5

**Summary:**

This article presents a special mapping method to encode secret information into the latent z, and then generates stego images using an existing diffusion model. The article is well-structured and easy to understand. However, in my opinion, the main contribution of this article lies in the three mapping methods, and there is relatively less innovation in this regard.

**Strengths:**

This article explores the impact of the distribution of latent z on the quality of generated images, which is indeed a question that exists and is worth investigating.

**Weaknesses:**

The key contribution of the article is the introduction of three mapping methods to embed secret information into the latent z. The image generation models are all pre-trained. I think the workload is relatively small, and there is limited innovation.

**Questions:**

1. The text mentions, "As generative steganography models do not have cover images, third-party players cannot train their steganalyzer models." When training a steganalyzer, real images are used as cover images, what about the stego images?Stego images are synthesized images containing secret information or real images with secret information? If stego images are synthesized images with secret information, how can third parties obtain them?
2. In the comparative experiments, both GSN and S2IRT use generated cover images and generated stego images to train the steganalyzer. What is the purpose of the author using real images for training?
3. Is there any theoretical derivation that can prove that the zm after message projection follows a Gaussian distribution, or is there any related experimental evidence?
4. In the experiments, in what format are the images generated by GSN and S2IRT saved, and has there been a standardized format? In reference [32], the images are saved in PNG format, why is it not used in the comparative experiments?
5. The author mentions, "The MC projection performs higher extracted message accuracy than the MN projection and better sample quality than the MB projection." The MC mapping method produces better image quality, but when generating stego images with the stable diffusion, why use the MB mapping method instead of the MC mapping?

**Details Of Ethics Concerns:**

None.

---

> ### Author Response · Authors · 2023-11-15
>
> Thank you for your comments about our paper.
>
> # Response to Weakness.
>
> In our method, two players only need to share the same generative model and the method of message projection.
> While it is possible to train new networks for message mapping, this approach significantly increases the amount of information that needs to be shared between the two participants.
> It also imposes an additional burden of learning new models, aside from diffusion models. Due to these considerations, we propose our training-free message projection, which is free from information-sharing requirements.
>
> # Response to Question 1.
>
> The mentioned stego images are indeed synthesized.
> It is challenging for the defender to obtain these stego images.
> The assumption in our experiment is that the defender somehow manages to acquire stego images and subsequently trains the steganalyzer.
>
> # Response to Question 2.
>
> In the context of generative steganography, we believe that a robust model should be capable of generating stego images that are indistinguishable from real images.
> For instance, models like GSN, which generate both stego and non-stego images, could be defended against by the adversary.
> After detecting the images generated by the model, the defender could employ techniques to remove the encryption, irrespective of whether the image is stego or non-stego.
>
> # Response to Question 3.
>
> Except for the MN methodology, $z_m$ does not follow the Gaussian distribution.
> However, since our goal is to create high-quality steganography using pre-trained diffusion models, we have devised a Gaussian distortion method that minimizes the degradation of image quality, as shown in figure 3,6,7, and 8.
> While MB distorts the distribution the most, it offers significant advantages in terms of accuracy.
>
> # Response to Question 4.
>
> All images generated by GSN and S2IRT are saved in the PNG format, consistent with our approach.
> As for reference [32] you mentioned, I'm not sure which specific reference it refers to.
>
> # Response to Question 5.
>
> We assume that the stable diffusion experiment you mentioned refers to a high-resolution EDM experiment.
>
> While the exact reason remains uncertain, one possible speculated reason is that the high-resolution model may not be well-trained for inputs that deviate significantly from Gaussian noise.
>
> Diffusion models are trained to denoise random noise during the learning process.
> Therefore, distorted $z_m$ through the MB or MC method may exhibit a noise distribution different from what the model learns to denoise.
> Consequently, denoising truncation errors in processing such images might be more pronounced compared to a low-resolution model.
> This could lead to difficulties in accurately restoring messages during the subsequent message extracting process, potentially due to excessively large truncation errors.
> The MB method, by maintaining a greater distance between two points, preserves accuracy well, whereas the MC method, with a shorter distance, might not perform as effectively.
>
> It's worth noting that the EDM proposed in our method primarily targets low-resolution images.
> Training EDM models on high-resolution images introduces potential instability during learning and might be a contributing factor to the observed decrease in the quality of the MC method.

---

### Official Review · Reviewer_Yv3H · 2023-10-31

**Soundness:** 2 fair
**Presentation:** 2 fair
**Contribution:** 1 poor
**Rating:** 3
**Confidence:** 4

**Summary:**

This paper proposes to use diffusion model for generative steganography, where three message projection strategies are proposed to encode the secret into Gaussian noise to fit the diffusion process.

**Strengths:**

1. The use of diffision model for generative image steganography.
2. The authors propose three strategies to map the secret into the Gaussion noises.

**Weaknesses:**

1. Limited technical contribution. Apart from the usage of the diffusion model, the only technical part is the message mapping strategy, which is rather limited and does not involved any learning process.

2. Experimental issue. There have been studies showing that, once the training images are available, it is farely easy to training a classifier to distinguish the diffusion model generated images from the real images[1]. However, according to the reported results, it seems that the classifier can hardly spot out the diffusion model generated images from the real images. As a matter of fact, in generative image steganography, a common knowledge is that it is almost impossible to generate stego-images with the same distribution as the real images. This is why we always evaluate the undetectability of the stego-images in terms of AIGC images without secret vs. stego-AIGC images.
[1] Zhu et al, GenImage: A Million-Scale Benchmark for Detecting AI-Generated Image, arXiv 2023.

3. Limited application value. Most of the images transmitted over the internet are in compressed format like the JPEG. As a matter of fact, most of the social network platforms do not allow the uploading of uncompressed images like the PNG. Therefore, the application value of such a scheme is rather limited.

4. Issues of illigal data extraction. Anyone who knows the mapping machenism will be able to do the message recovery, isn't it?

**Questions:**

See weakness

---

> ### Author Response · Authors · 2023-11-15
>
> Thank you for your comments about our paper.
>
> # Response to Weakness 1.
> We did not introduce additional technical sources such as a learning process.
> The reason behind this decision is that incorporating elements like a learning process would increase the amount of information that the sender and receiver need to be aware of.
> If we were to train a new network for use in the projection, both players would have to share the additional weights of the network.
> Moreover, introducing training would entail extra computational costs in the process.
> Due to these considerations, we propose our training-free message projection, which is free from information-sharing requirements.
>
>
> # Response to Weakness 2.
>
> [1] suggests that the distinguishability between generated and real images is contingent upon the dataset's size, which, in the proposed study, is extensive, comprising 1.3 million images. Notably, the effectiveness of discrimination is shown to decline when the dataset is limited, emphasizing the pivotal role of dataset size.
> In our experiments, the steganalyzers are trained with 5,000 real images, potentially leading to scenarios where the model fails to discern whether an image is a stego image or not, especially if the image quality is sufficiently high.
>
> # Response to Weakness 3.
>
> That's true, but considering formats like PNG, which offer a reasonable balance between quality and file size compared to high-capacity formats like TIFF, we believe there's still practical value.
> Overcoming this limitation could be a subject for future research.
>
> # Response to Weakness 4.
>
> To decipher the message embedded in our steganography method, the receiver needs to possess information about the message projection method and the parameters of the diffusion model. If third-party players are unaware of the specific model used to generate the image, even if they understand the mapping mechanism, they cannot recover the message.

---

> > ### Comment · Reviewer_Yv3H · 2023-11-23
> > **Comments from Reviewer Yv3H**
> >
> > I maintain my rating after reading the rebuttal. Note that, according to Kerckhoffs's principle, we should assume that the attacker knows the algorithm  to analyze the security.

---

### Official Review · Reviewer_Ny5h · 2023-11-09

**Soundness:** 3 good
**Presentation:** 3 good
**Contribution:** 2 fair
**Rating:** 1
**Confidence:** 4

**Summary:**

A diffusion based generative steganography method is presented. Secrete messages are projected into the noise of the diffusion model and produces stego images during the reverse process. The projection trades off between message accuracy, antidetection ability and quality of the image. The proposed method generates high quality stego image with high capacity message.

**Strengths:**

-The readability of the paper is good.
- The proposed diffusion based generative steganography produces high fidelity stego images with higher capacity message than compared methods.

**Weaknesses:**

-The high stego image quality is achieved not by any novelty but it comes from the diffusion model itself. In addition, the other properties that you look for in a stego image probably comes from the diffusion process and not from any novelty that the paper proposes.

-Although the projection is a way for introducing message into the diffusion model but it would be difficult to prove that it is a mathematically optimum way of introducing the message. Multiple bits can be introduced in the projection. The paper discusses two bits but you can generalize to more than two bits. It would be interesting to know their performance.

-This paper provides one working method for producing stego images but it lack mathematical analysis or theory. The paper is suggesting one engineering method for generating stego images based on the diffusion process and it is difficult to say it is anyway unique.

- How do you deal with the scalability issues with the proposed method?

- What can you say about the computation and latency issue with the proposed method?

**Questions:**

Please see weakness.

---

> ### Author Response · Authors · 2023-11-15
>
> Thank you for your comments about our paper.
> # Response to Weakness 1.
>
> We acknowledge the novelty of proposing a method that capitalizes on the characteristics of previously suggested models to undertake a distinct task, such as steganography.
>
> As shown in Figures 3,6,7, and 8, we noted a degradation in image quality and the incapacity to generate images when manipulating the noise of diffusion models. Consequently, we introduced message projection as a means of manipulating the noise using diffusion models for steganography, which, we believe, introduces novelty.
>
> This novelty extends to the features obtained from stego images. Our method not only ensures higher image quality compared to other existing generative steganography approaches but also facilitates the use of the proposed novelty, message projection, without significantly impacting the model's training or the generation process of the trained model. Furthermore, the accuracy of our model, manipulated by the noise through our message projection, allows us to adjust the accuracy-quality trade-off, addressing issues that could arise from naive methods, as demonstrated in Appendix Figures 6,7, and 8.
>
> Rather than treating diffusion models as a black box, we introduced a new method by combining the mathematical properties proposed in previous diffusion model research [1, 2] and our experimental analysis. Considering these factors, it is challenging to argue that there is no novelty in our proposal.
>
> # Response to Weakness 2.
>
> We evaluate the message accuracy using 3 bpp, 6 bpp,9 bpp and 12 bpp messages, considering 1,000 stego images with a pre-trained EDM on the AFHQv2 dataset.
> |Capacity|Accuracy|
> |-|-|
> |3 bpp | 99.37 |
> |6 bpp | 91.56 |
> |9 bpp | 83.18 |
> |12 bpp | 60.40
>
> # Response to Weakness 3.
>
> We believe that we are adequately following the mathematical analysis of existing diffusion models. Through previous research on diffusion models, the image generation process is represented by stochastic differential equations (sde) or ordinary differential equations (ODE). Consequently, truncation errors occur during the generation process due to these characteristics. Additionally, data loss occurs during the image storage process. These outcomes result in errors when utilizing diffusion models for steganography. Therefore, to overcome these errors, we propose message projection, based on our analysis, we do not consider our method, including message projection, to be unique.
>
> A1. Our approach inherits the computational characteristics of the underlying diffusion models, as we directly leverage existing diffusion models.
> Moreover, the algorithm implemented in our method exhibits a time complexity of $O(N)$, with $N$ representing the number of binary messages.
>
> A2. Compared to DDPM, recent diffusion models demand a lower computational cost for image generation. Moreover, various methods[3,4] have been suggested to address the latency issue associated with them. We demonstrated the applicability of our approach to recently proposed various diffusion models, and we anticipate its relevance to future research. While the models we employed in this paper are already adept at mitigating this issue, we believe that further research will provide additional solutions.
>
> [1] Karras, Tero, et al. "Elucidating the design space of diffusion-based generative models." Advances in Neural Information Processing Systems 35 (2022): 26565-26577.
>
> [2] Song, Yang, et al. "Score-based generative modeling through stochastic differential equations." In the 9th International Conference on Learning Representations, 2021.
>
> [3] Vahdat, Arash, Karsten Kreis, and Jan Kautz. "Score-based generative modeling in latent space." Advances in Neural Information Processing Systems 34 (2021): 11287-11302.
>
> [4] Rombach, Robin, et al. "High-resolution image synthesis with latent diffusion models." Proceedings of the IEEE/CVF conference on computer vision and pattern recognition. 2022.